# Proteomic Analysis of Salivary Secretions from the Tea Green Leafhopper, *Empoasca flavescens* Fabrecius

**DOI:** 10.3390/insects15040296

**Published:** 2024-04-22

**Authors:** Cheng Pan, Xueyi He, Luxia Xia, Kexin Wei, Yuqun Niu, Baoyu Han

**Affiliations:** Zhejiang Provincial Key Laboratory of Biometrology and Inspection & Quarantine, China Jiliang University, Hangzhou 310018, China; panchengtt@163.com (C.P.); hexy1031@163.com (X.H.); xialuxia1229@163.com (L.X.); wkxxxxkw@163.com (K.W.); hanby15@163.com (B.H.)

**Keywords:** tea green leafhopper, salivary proteins, *Empoasca flavescens*, gene cloning, expression pattern

## Abstract

**Simple Summary:**

The tea green leafhopper, *Empoasca flavescens* Fabrecius, is the most notorious pest in Chinese tea plantations, predominantly controlled through pesticide applications for decades. Recognizing the pressing need for novel, effective, and eco-friendly strategies to manage leafhoppers, this study aims to contribute to the reduction in pesticide usage in the tea industry. Salivary protein research is pivotal for gaining insights into effectors or elicitors, facilitating the exploration of novel regulatory targets and pathways in tea green leafhoppers. In this study, saliva from adult tea green leafhoppers was collected using a self-made collection device equipped with two layers of Parafilm stretched over a sucrose diet. Following freeze drying and filter-aided sample preparation (FASP), liquid chromatography–tandem mass spectrometry (LC-MS/MS) was employed to analyze the saliva proteins. Bioinformatics methodologies were then leveraged to predict the secretory proteins. The results suggest that the interactions between leafhoppers and tea plants, particularly in terms of feeding and defense responses, may be mediated by elicitors or effectors, such as mucin-like protein (EfMucin), vitellogenin (EfVg), odorant-binding protein (EfOBP), and others. The conclusive outcomes of this study provide new insights into the coevolutionary dynamics between tea plants and leafhoppers, offering novel pathways for the development of advanced leafhopper control technologies.

**Abstract:**

Saliva plays a crucial role in shaping the compatibility of piercing–sucking insects with their host plants. Understanding the complex composition of leafhopper saliva is important for developing effective and eco-friendly control strategies for the tea green leafhopper, *Empoasca flavescens* Fabrecius, a major piercing–sucking pest in Chinese tea plantations. This study explored the saliva proteins of tea green leafhopper adults using a custom collection device, consisting of two layers of Parafilm stretched over a sucrose diet. A total of 152 proteins were identified using liquid chromatography–tandem mass spectrometry (LC-MS/MS) following the filter-aided sample preparation (FASP). These proteins were categorized into six groups based on their functions, including enzymes, transport proteins, regulatory proteins, cell structure proteins, other proteins, and unknown proteins. Bioinformatics analyses predicted 16 secreted proteins, which were successfully cloned and transcriptionally analyzed across various tissues and developmental stages. Genes encoding putative salivary secretory proteins, including *Efmucin1*, *EfOBP1*, *EfOBP2*, *EfOBP3*, *Efmucin2*, low-density lipoprotein receptor-related protein (*EfLRP*), *EFVg1*, and *EFVg2*, exhibited high expressions in salivary gland (SG) tissues and feeding-associated expressions at different developmental stages. These findings shed light on the potential elicitors or effectors mediating the leafhopper feeding and defense responses in tea plants, providing insights into the coevolution of tea plants and leafhoppers. The study’s conclusions open avenues for the development of innovative leafhopper control technologies that reduce the reliance on pesticides in the tea industry.

## 1. Introduction

Piercing–sucking insects, such as leafhoppers, planthoppers, and aphids, are notorious pests that cause damage to host plants through feeding, spawning, or virus transmission [1,2,3,4,5]. The mouthpart serves as a crucial organ for insects to acquire nutrients, with the stylet being a unique feeding structure evolved through the coevolutionary process between piercing–sucking insects and plants [2,3]. This organ not only facilitates puncturing but also minimizes exposure to defense substances on plant surfaces, making it easier for insects to extract plant juices for nutrition [6,7]. As feeding strategies evolve, the physiological and biochemical functions of insects undergo necessary adjustments. Saliva, as a fundamental substance in the interaction between insects and plants, plays a complex and important role in this relationship [7,8,9]. Recent studies on the saliva of piercing–sucking insects have unveiled the essential roles played by the salivary components, originating from the salivary duct (salivary gland (SG)) or food duct (intestinal tract) connected to the stylet, in the relationships between these insects and plants [3,7,9]. Interestingly, some salivary components are considered elicitors capable of triggering plant defenses, while others inhibit elicitor-induced plant defenses [3,7,9]. However, the intricate interactions within saliva remain poorly understood. Consequently, research on the saliva components of piercing–sucking insects holds the potential to provide valuable insights for effective pest management, making their control strategies a consistently hot research topic.

The stylet of piercing–sucking insects is responsible for penetrating both the phloem and xylem of a plant for feeding. During the feeding process, two types of saliva are ejected into plant tissues [3]. The first type, known as gel saliva, is secreted during stylet probing, providing mechanical stability and lubrication for stylet movement [3,8,9,10,11]. The salivary sheath formed by gel seals the stylet penetration site, preventing plant immunity triggered by leaked cell components [9,10,11]. Disrupted salivary sheath formation in aphids and planthoppers can hinder insect feeding from plant sieve tubes, but not from artificial diets [12,13,14]. Watery saliva, the second type, is secreted by the insect as it pierces plant tissue and contains various components, such as pectinase, cellulase, polyphenol oxidase, and sucrase. These components aid in puncturing plants, digesting food, detoxifying secondary substances, and disrupting plant defense responses [3,8,11]. Moreover, the components present in saliva can stimulate plant defense responses. This includes triggering the generation of injury signals, constituting direct defense, and prompting plants to release volatiles, attracting predators as part of indirect defense. Specific components found in insect watery saliva act as elicitors, inducing plant defense [7,9]. The application of synthesized elicitor analogs before pest outbreaks has the dual effect of not only prompting plants to produce direct defense substances but also activating natural enemies to regulate insect populations through predation or parasitism. This offers theoretical guidance for the development of environmentally friendly management strategies.

Hence, the prerequisite for understanding the role of saliva in inducing plant defense lies in the identification of the salivary components, particularly protein. Recent advancements in nucleic acid-sequencing technology and proteomics have greatly propelled research on insect salivary proteins. SG transcriptome sequencing has been a valuable tool for predicting and understanding the composition of salivary proteins. Noteworthy studies in recent years have focused on the transcriptome-sequencing analysis of Hemiptera insect SGs, such as 295 proteins in *Bemisia tabaci* SGs in 2012 [15], 163 in *Macrosiphum euphorbiae* in 2013 [16], 526 in *Sitobion avenae* (wheat aphid) in 2017 [17], and 3603 in *Acyrthosiphon pisum* (pea aphid) in 2018 [18]. This highlights the efficiency of transcriptome analysis in rapidly predicting the protein composition of SGs. However, discerning actual salivary secretion proteins from proteins of the SGs remains challenging. Scholars have addressed this challenge by utilizing liquid chromatography–tandem mass spectrometry (LC-MS/MS) to directly analyze the protein composition in saliva. For instance, Chaudhary et al. [19] employed this method to predict 102 salivary proteins in the saliva of adult potato aphids (*M. euphorbiae*), collected using a device with two layers of Parafilm and a resorcinol diet designed to stimulate saliva secretion. This collection and identification method, widely applied in aphids, identified 34, 32, 12, 7, and 87 salivary proteins from *Diuraphis noxia*, *Schizaphis graminum*, *S. avenae*, *Metopolophium dirhodum*, and *Megoura viciae*, respectively [20,21,22,23], showcasing its versatility. In summary, SG transcriptome or salivary proteome analysis in piercing–sucking insects proves to be an effective approach for obtaining highly reliable salivary proteins. The combined analysis of transcriptome and proteome data enhances data corroboration, providing a more detailed understanding of the salivary protein secretion spectrum.

Over the decades, concerns about tea quality and safety have escalated due to extensive pesticide applications aimed at controlling tea green leafhoppers, *Empoasca flavescens*. Moreover, the resistance of leafhoppers, particularly to bifenthrin and acetamiprid, has risen significantly, reaching an unacceptable level. Limited studies have been conducted on the salivary secretions from tea green leafhoppers. Similar to most other Hemiptera insects, the saliva of tea green leafhoppers during the feeding process contains various components, including enzymes like laccase-1 and *β*-glucosidase [24,25,26], along with a Ca^2+^-binding protein [27]. Consequently, the study of salivary secretion holds promise for generating novel, effective, and environmentally sound control measures against leafhoppers, aiming to reduce or eliminate the need for pesticides in the tea industry.

In this study, saliva from adult tea green leafhoppers was collected using a self-made collection device involving two layers of Parafilm and a sucrose diet. Following concentration through freeze drying, the saliva protein solution underwent hydrolysis through the filter-aided sample preparation (FASP) method. Subsequently, the saliva proteins were analyzed using LC-MS/MS. The identification of salivary proteins was integrated with the genome and transcriptome of *E. flavescens*. The study investigated the specific expression patterns of predicted secretory proteins, analyzing their potential involvement in tea plant defense. Our study provides an encompassing overview of *E. flavescens* salivary proteins, shedding light on the interaction mechanism between tea green leafhoppers and tea plants. Furthermore, our attempt to identify candidate effectors in *E. flavescens* opens new possibilities for pest management strategies.

## 2. Materials and Methods

### 2.1. Insects

The populations of the tea green leafhopper, *E. flavescens*, were sourced from Langxi County, Xuancheng City, Anhui Province, in May 2019. Field studies did not involve any endangered or protected species, and no specific permissions were required for these activities at this station. The insects were bred on the tea cultivar “Wuniuzao” (*Camellia sinensis* cv. Wuniuzao) and were housed in insect cages within a pesticide-free greenhouse, maintaining conditions at 25 ± 2 °C and 70 ± 5% relative humidity under a 14:10 (light:dark) photoperiod.

Tissues, including SGs, integuments, fatty bodies, guts, testes, and ovaries, were dissected from newly emerged adult (less than 24 h old) tea green leafhoppers using a dissecting microscope (Leica Microsystems GmbH, Wetzlar, Germany) and microforceps (Shanghai Medical Instruments Ltd., Corp., Shanghai, China) in chilled 1× Phosphate-Buffered Saline (1×PBS) solution (pH 7.2, Life Technologies Corporation, Carlsbad, USA). Different developmental stages of tea green leafhoppers, comprising eggs, the 1st–5th instar nymphs, and newly emerged female and male adults (less than 24 h old) were systematically collected. Three independent biological replicates of each sample were collected. Immediately upon collection, the samples were treated with liquid nitrogen and stored at −80 °C for subsequent RNA extraction.

### 2.2. Collection, Concentration, and Digestion of Watery Saliva

Watery saliva was collected by feeding tea green leafhoppers at various developmental stages in a self-made collection container. The container, consisting of two layers of Parafilm stretched with a 3000 μL 5% sucrose diet, was employed following methods similar to those used for aphids [19], brown planthoppers [14], and white-backed planthoppers [5], with certain modifications [28] (Appendix A). Each saliva collection container was positioned on one side of a sterile glass tube (5.5 cm × 10 cm), with an additional layer of Parafilm on the outermost side to prevent leakage. The opposite end was used to introduce the leafhoppers. After introducing the leafhoppers, the open end of the glass tube was secured with cheesecloth and a rubber band. The glass tubes were placed in chambers maintained at 25 ± 2 °C and 70 ± 5% relative humidity under a 14:10 (light period:dark period) photoperiod. All components were either sterilized or treated with alcohol. After 24 h, the saliva-containing diet was collected under sterile conditions using an injector needle, pouring the stylet-probed diet into sterile centrifuge tubes with a pipette. With twenty leafhoppers in each tube and no leafhoppers reused for saliva collection, a total of 8000 individual leafhoppers were collected as one sample. Three independent biological replicates were harvested for each variety. The collected saliva-containing diet was stored at −80 °C.

The collected saliva-containing diet underwent a freeze-drying process, involving freezing at −80 °C for 120 h, followed by dehydration in a freeze dryer under vacuum (0.011 mbar) at −60 °C. The dried sample was dissolved in 1 mL of deionized water. Subsequently, the sample was dissolved in SDT buffer consisting of 4% sodium dodecyl sulfate (SDS) (Bio-Rad, Hercules, CA, USA), 100 mM Tris-HCl, and 1 mM dithiothreitol (DTT) (Bio-Rad, Hercules, CA, USA) at pH 7.6, incubated in hot water for 10 min, and centrifuged at 14,000× *g* for 30 min. The protein samples were quantified by the BCA method, sub-packaged, and stored at −80 °C.

For protein digestion, the FASP method [29] was employed. A 100 μL protein sample was added to 100 mM DTT and incubated in hot water for 5 min. After adding 200 μL UA buffer (8M urea (Bio-Rad, Hercules, CA, USA), 150 mM Tris-HCl, pH 8.0), the mixture was transferred to a 10 kDa ultrafiltration centrifuge tube (Sartorius, Gottingen, Germany) and centrifuged twice at 14,000× *g* for 15 min. This step was repeated once. A 100 μL IAA (Iodoacetamide) buffer (100 mM IAA in UA) was added to the tube, incubated in the dark for 30 min at room temperature after vortexing for 1 min at 600 rpm, and then centrifuged at 14,000× *g* for 15 min to remove filtrate. The sample was then added to 100 μL UA buffer and centrifuged at 14,000× *g* for 15 min to remove filtrate. This procedure was repeated once. The sample was digested with 40 μL trypsin buffer (2 μg trypsin in 40 μL 100 mM NH_4_HCO_3_), reacting for 16–18 h at 37 °C after vortexing for 1 min at 600 rpm. The sample was centrifuged at 14,000× *g* for 15 min and transferred to a new collection tube. An amount of 40 μL 25 mM NH_4_HCO_3_ was added to the tube and then centrifuged at 14,000× *g* for 15 min to collect filtrate. A C18 Cartridge (Sigma, St. Louis, MO, USA) was used to desalinate the peptides. After lyophilizing the peptides, 40 μL 0.1% formic acid (FA) (Sigma, St. Louis, MO, USA) was added. Finally, peptides were qualified at OD_280_.

### 2.3. LC-MS/MS Analysis of Watery-Saliva Proteins

The LC/MS-MS analysis was conducted using an Easy nLC system (Thermo Fisher Scientific, Waltham, USA) coupled with a Q Exactive mass spectrometer (Thermo Fisher Scientific). For the analysis, 6 μL digested peptides were utilized. The mobile phases consisted of 0.1% formic acid aqueous solution for phase A and a mixture of 0.1% formic acid and 84% acetonitrile (ACN) (Merck, Darmstadt, Germany) aqueous solution for phase B. Peptides were loaded onto a C18 reversed-phase column (Thermo Scientific EASY column, 100 μm × 2 cm, 5 μm-C18) and separated in an analytical column (Thermo scientific EASY column, 75 μm × 10 cm, 3 μm-C18) at a flow rate of 300 nL/min. The liquid-phase gradient was set as follows: from 0% B to 55% B over 0–110 min, from 55% B to 100% B over 110–115 min, and 100% B at 115–120 min.

Peptide separations were analyzed using a Q Exactive mass spectrometer (Thermo Fisher Scientific, MA, USA) by dynamically choosing the 20 most abundant ions from one full mass scan (300–1800 *m*/*z*) for high-energy collisional dissociation (HCD) fragmentation. The normalized collision energy was set at 27 eV, and dynamic exclusion was applied for 60 s. The underfill ratio was defined as 0.1%. The resolution of the first level of the mass spectrum was set at 70,000 at *m*/*z* 200, while the second level was set at 17,500 at *m*/*z* 200. The automatic-gain-control (AGC) target was set to 3e6.

### 2.4. Protein Identification

The raw files underwent a thorough search using Maxquant software (Max Planck Institute of Biochemistry in Martinsried, Germany) against three databases: (a) the public Uniprot database tailored for Auchenorrhyncha (http://www.uniprot.org, 182,925 putative proteins (accessed on 12 June 2020), (b) the Genome database specific to the tea green leafhopper (genome assembly ASM1883171v1), and (c) the transcriptomic database specific to the tea green leafhopper (unpublished). For protein identification, the search parameters were configured as follows: (a) enzyme: trypsin; (b) max. missed cleavages: 2; (c) fixed modifications: carbamidomethyl (C); (d) variable modifications: oxidation (M), Acetyl (protein N-terminal); and (e) Mascot score: greater than 20.

### 2.5. Bioinformatics Analysis

The identification outcomes of tea green leafhopper salivary proteins were validated through the Swiss-Prot/Tr EMBL database (http://www.uniprot.org/ (accessed on 15 March 2021) [30]. Additionally, Pfam version 31.0 (https://www.ebi.ac.uk/interpro/ (accessed on 20 March 2021) [31] and other databases were employed to ascertain their structure domains, functions, and other pertinent information. Gene Ontology (GO) annotations for salivary proteins were conducted (http://geneontology.org/ (accessed on 20 March 2021) [32]. The Blast KOALA program of the Kyoto Encyclopedia of Genes and Genomes (KEGG) database was used to annotate saliva proteins within the signal pathways of the KEGG PATHWAY database (http://www.kegg.jp/ (accessed on 20 March 2021) [33]. For the prediction of protein subcellular localization, the Target P 1.1 Server (http://www.cbs.dtu.dk/services/TargetP/ (accessed on 25 March 2021) was utilized [34]. Salivary protein signal peptide prediction was executed using the Signal P 4.1 Server (http://www.cbs.dtu.dk/services/SignalP/ (accessed on 25 March 2021) [35], with an analysis of the signal peptide’s positions and cleavage sites. The prediction of transmembrane regions was accomplished through the THMHH Server v2.0 (http://www.cbs.dtu.dk/services/TMHMM/ (accessed on 25 March 2021) [36].

### 2.6. Prediction and Cloning of Genes Encoding Salivary Secretory Proteins

To predict the amino acid sequences of secreted protein, adherence to the criteria outlined in [35,37,38] was essential: (a) the presence of a signal peptide and the absence of a transmembrane domain; (b) the presence of both a signal peptide and a transmembrane domain, with the latter falling within the range of the signal peptide. Using the peptide and protein sequences of potential secretory proteins as a “Query”, searches were conducted in both the Genome database and transcriptomic databases of the tea green leafhopper. The strategy and procedural steps for acquiring each gene of salivary protein mirrored those outlined by Pan et al. [39].

Specific primer pairs (Appendix A) for cloning the full-length cDNA sequences were designed using Primer Premier 5 (Premier Biosoft International, Palo Alto, CA, USA) and synthesized by Sangon Biotech Co. (Shanghai, China). The SMART RACE cDNA Amplification Kit (Clontech, Mountain View, CA, USA) was employed to isolate full-length cDNAs following the manufacturer’s instructions [40]. The resulting PCR products were purified using the Axygen Gel Extraction Kit (Axygen, Union City, CA, USA), and subsequent ligation into the pMD-20T vector (TAKARA, Tokyo, Japan) was carried out. Plasmid constructs were sequenced by Sangon Biotech Co., with the sequence accuracy verified through sequencing.

### 2.7. Gene Expression Analysis through qRT-PCR

For quantitative analysis of the gene expression, the StepOnePlus™ real-time PCR system (Thermo Fisher Scientific, Foster City, CA, USA) was used. Before analysis, ROX reference dye was combined with TB SYBR Premix Ex Taq II (Tli RNase H Plus, TAKARA) to normalize signals and ensure data integrity. Total RNA was extracted from samples using TRIzol Reagent (Invitrogen, MA, USA). Reference genes, *EfTubulin* and *EfActin*, were used to normalize the expressions of interesting genes across various stages and tissues [41].

Specific qRT-PCR primers (Appendix A) were designed using Primer Premier 5 and synthesized by Sangon Biotech Co. (Shanghai, China). Primer specificity and efficiency were confirmed by analyzing standard curves with a tenfold cDNA dilution series. The qRT-PCR reaction volume was 25 μL, comprising 12.5 μL of SYBR Premix Ex Taq II (Tli RNase H Plus), 1 μL of forward primer, 1 μL of reverse primer, 2 μL of cDNA template, and the addition of ddH_2_O to reach a 25 μL volume. The reaction program involved initial denaturation at 95 °C for 30 s, followed by 40 cycles at 94 °C for 5 s and 60 °C for 30 s. Three independent qRT-PCR experiments were performed on total RNA samples. Mean and standard deviation values were calculated based on three independent biological replicates.

Relative gene expression levels were determined using the comparative 2^−ΔΔCt^ method [42]. Regarding the histogram, the 2^−ΔΔCt^ values of the transcripts of the putative salivary secretory protein genes from various tea green leafhopper tissues samples were calculated and expressed as the fold change relative to the expression of the SG tissues (expression = 1). In the different developmental-stage samples, the values were calculated and expressed as the fold change relative to the expression of the 1st instar nymphs (expression = 1). Significant differences in gene expression levels were identified using Tukey’s multiple-range test at a 0.05 probability level.

## 3. Results

### 3.1. Identification of Saliva Proteins

Employing a shotgun combined with the LC-MS/MS analysis method, a total of 152 proteins were identified from the watery saliva of tea green leafhoppers (Appendix A). The categorization of these proteins was based on their functions, domains, and functional annotations, leading to six distinct categories: (a) enzymes, encompassing hydrolases, oxidoreductases, ligases, translocases, transferases, lyases, and isomerases; (b) transport proteins, inclusive of lipid transporters, ion transporters, endoplasmic reticulum (ER)–Golgi transporters, ABC transporters, and vacuolar transporters; (c) regulatory proteins, comprising transcription factors and binding proteins involved in DNA, RNA, and protein interactions; (d) cell structure proteins, including cytoskeleton proteins, actins, tubulins, ribosomal proteins, and spliceosome components; (e) other proteins, including ubiquitin proteins, molecular chaperones, heat shock proteins, and signal transduction proteins; and (f) unknown proteins.

GO annotation analysis was conducted on the identified 152 tea leafhopper salivary proteins. These proteins were classified at the second level under three main GO domains: molecular functions, biological processes, and cellular components. The results, presented in Figure 1, highlighted binding (50 proteins) and catalytic activity (22 proteins) as the top two molecular functions. Within biological processes, the major categories were the cellular process (39 proteins), metabolic process (22 proteins), and single-organism process (19 proteins). Regarding cellular components, cells (32 proteins), cell components (31 proteins), and organelles (25 proteins) were the most prevalent locations.

Furthermore, salivary proteins were annotated in the KEGG pathways for a comprehensive understanding of their functions. The KEGG pathways related to salivary proteins were categorized into five branches: metabolism, genetic information processing, environmental information processing, cellular processes, and organismal systems (Figure 2). At the first level, metabolism (38 proteins), organismal systems (36 proteins), and cellular processes (27 proteins) stood out as the top three categories, with global and overview maps (24 proteins) and signal transduction (20 proteins) being the most frequent categories at the second level.

### 3.2. Predicting and Cloning of Genes Encoding Putative Salivary Secretory Proteins

Out of 152 distinct salivary proteins, we identified 16 potential secretory proteins in the predictive analysis of THMHH and SignalP. This subset comprises 13 classic secretory proteins, characterized by both a transmembrane domain and a signal peptide, along with three additional proteins possessing a single transmembrane domain. Among these, vitellogenin (Vg), pyrroline-5-carboxylate reductase (P5CR), s-adenosyl-L-methionine-dependent methyltransferase, odorant-binding protein (OBP), and 12 unknown proteins were identified. Notably, the functions of these 12 unidentified proteins remain unknown or are absent within Auchenorrhyncha, rendering them distinct from the tea green leafhopper. Upon scrutinizing these putative secretory proteins in the tea green leafhopper transcriptome, 20 proteins displayed partial open reading frames (ORFs), indicating their reliability for sequencing and subsequent cloning. Conversely, other proteins either eluded detection in the transcriptome or exhibited only truncated fragments, leading to their exclusion from further analysis. The complete ORFs of the 16 identified proteins were validated through the cloning and sequencing process, while partial ORFs of the remaining four proteins were successfully obtained. Subsequently, all 20 putative secreted proteins underwent manual scrutiny via the Blastx program and were named based on the highest protein similarity documented in the National Biotechnology Information Center (NCBI). The login numbers and related information for each protein are delineated in Table 1.

### 3.3. Tissue-Specific Expressions of Putative Salivary Secretory Protein Genes

Salivary proteins play a pivotal role in mediating interactions between insects and plants, predominantly originating from either the salivary duct or food duct (intestinal tract) [3,7,9]. The tissue-specific expression patterns of genes encoding secretory putative proteins are closely associated with their sources and physiological functions. Therefore, a comprehensive analysis of the target gene expression across specific tissues and organs is imperative.

The abundances of the transcripts of twenty secretory putative protein genes were determined in five different SG tissues: integument (IN), fatty body (FB), gut (Gut), testis (TE), and ovary (OV) tissues. Employing *EfTubulin* and *EfActin* as internal reference genes revealed their ubiquitous expressions across all tissues in spite of the varying transcript levels (Figure 3). Notably, secretory putative protein genes, such as *Efmucin1*, *EfOBP1*, *EfOBP2*, *EfOBP3*, *Efmucin2*, *EfLRP*, *EFVg1*, and *EFVg2*, demonstrated peak expression levels in SG tissues. Specifically, *Efmucin1*, *EfOBP1*, *EfOBP2*, and *EfOBP3* exhibited expression levels over 5 times higher than those in other tissues. These genes, showing heightened expression in SG or gut tissues, are likely intricately linked to feeding processes and merit specialized attention. However, certain secretory putative protein genes (*EfP5CR*, *EfUP3*, *EfMME*, and *EfNUP*) displayed consistent expression levels across tissues. Notably, eight secretory putative protein genes (*EfelF3*, *EfCCDC*, *EfRGN*, *EfUP2*, *EfVDCC*, *EfE3UPL*, *EfTTP*, and *EfUP1*) exhibited lower expression levels in SG tissues compared to other tissues. The correlation of this phenomenon with food intake remains to be elucidated, necessitating further investigation.

### 3.4. Development-Specific Expressions of Putative Salivary Secretory Protein Genes

Studying the expression patterns across various growth and development phases is crucial due to variations in Hemiptera insect feeding habits at different life stages and the distinctive dietary preferences of female insects before mating [7,8,9,14]. The expression profiles of twenty putative salivary secretory protein genes were assessed in different developmental stages and sexes, encompassing eggs, the first–fifth instar nymphs, and newly emerged female and male adults (less than 24 h old), using qPCR (Figure 4). The expression level of the first instar nymphs served as the baseline, set as 1. As the insects progressed through their life stages, the expressions of three genes (*EfVg1*, *EfVg2*, and *Efmucin1*) increased, reaching their peaks in either male or female adults. This aligns with the expected characteristics of genes associated with feeding behavior. Conversely, the majority of secretory protein genes, including *Efmucin2*, *EfE3UPL*, *EfVDCC*, *EfelF3*, *EfCCDC*, *EfLRP*, *EfMME*, *EfNUP*, *EfRGN*, and *EfOBP2*, showed increased expression levels between the first and third instar nymph stages, with comparatively lower levels in male or female adults. Notably, certain secretory protein genes, such as *EffTTP*, *EfOBP1*, *EfOBP3*, and *EfUP1*, reached peak expression during the first instar nymph stage, while *EfP5CR*, *EfUP2*, and *EfUP3* exhibited peak expression during the fifth instar nymph stage. This divergence in expression patterns highlights the dynamic regulatory mechanisms governing the developmental stages of these genes, contributing to the intricate life cycle of the tea green leafhopper.

## 4. Discussion

Recent technological advancements have spurred increased research into the salivary proteins of sucking insects, such as potato aphids [19], green peach aphids (*Myzus persicae*) [43], mirid bugs (*Apolygus lucorum*) [44], and whiteflies (*Aleyrodidae*) [45]. However, there is a notable gap in understanding the salivary components of Auchenorrhyncha species, particularly those that are phytophagous. To comprehensively grasp the fundamental roles of the salivary components in vascular bundle feeding, especially from the phloem, a comparative analysis of the Auchenorrhyncha and Sternorrhyncha salivary components is essential, involving leafhoppers and aphids, respectively.

Our investigation focused on the saliva components of *E. flavescens* to decipher the interaction between the insects and their host plants. Analyzing information about protein activities, domains, and functional annotations led to the identification of 152 proteins categorized into six groups. Recent findings on the watery saliva from brown planthoppers [14], white-backed planthoppers [5], and rice leafhoppers [46] indicated the presence of 206, 161, and 71 proteins, respectively. Upon comparison, relative consistency in the protein classification emerged, predominantly encompassing enzymes, transport proteins, regulatory proteins, cell structure proteins, and other categories. Furthermore, our study revealed overlaps with previous research, identifying dozens of salivary proteins shared with ligases, translocases, lyases, isomerases, heat shock proteins, lipid transporters, ion transporters, cytoskeleton proteins, actins, tubulins, and ribosomal protein. However, in contrast to our findings, over 60% of the 71 saliva proteins detected by rice leafhoppers were identified as functional proteins or enzymes, such as GH5 cellulase, transferrin, carbonic anhydrases, aminopeptidase, regucalcin (RGN), and apolipoprotein [4]. Although we identified certain enzymes, their types and concentrations were relatively low. A study conducted by Shao et al. [46], involving transcriptome and proteome studies on the SGs and intestines of the tea green leafhopper, indicated that the cysteine protease and the serine protease were the primary digesting enzymes in the intestines. However, our findings diverged from theirs, likely attributed to differences in the insect species, saliva collection methods, and LC/MS-MS analysis approaches.

Insect secretory proteins are considered potential effectors that are capable of being secreted into plant tissues, thereby modifying plant defense mechanisms [3,7,8,9]. These proteins released by insects are believed to play a crucial role in both defense responses and feeding processes [3,7,8,9]. Employing bioinformatics approaches, we predicted 16 peptide segments corresponding to secreted proteins among the 152 proteins that we identified. Subsequently, we identified and cloned 20 genes encoding secretory putative secretory proteins from the transcriptome of the tea green leafhopper. These genes included E3 ubiquitin-protein ligase (E3UPL), voltage-dependent calcium channel (VDCC), P5CR, eukaryotic translation initiation factor 3 (elF3), alpha-tocopherol transfer protein (TTP), coiled-coil domain-containing protein (CCDC), low-density lipoprotein receptor-related protein (LRP), membrane metallo-endopeptidase (MME), nuclear pore complex protein (NUP), RGN, Vg, mucin-like protein, and OBP. Similarly, Huang et al. [14] anticipated 19 genes in the saliva of brown planthoppers, which encode five membrane-related proteins, six SG proteins (with specific proteins remaining unknown), seven potential enzymes, and a mucin-like protein. In the case of white-backed planthoppers, Miao et al. [5] predicted 11 genes in the saliva, including peptidylglycine α-hydroxylating monooxygenase, carboxylesterase, neprilysin-11-like isoform X4, serine protease 6, carbonic anhydrase 2, protein disulfide-isomerase, vacuolar ATP synthase subunit E, lipophorin precursor, Vg, calexcitin-1 isoform X1, and mucin-like protein. Despite the commonality of utilizing identical artificial diets to obtain saliva proteins in these three studies, a comparison highlighted significant variations in the projected outcomes of the putative salivary secretory proteins. These differences may be attributed to species variations and differences in the feeding targets. Nevertheless, some similar proteins were also identified, such as enzymes, alpha-tocopherol transfer proteins, Calcium ion-related proteins, Vg, and mucin-like proteins [5,14]. Although we screened 16 putative secretory proteins from the repertoire of proteins for further investigation, relying solely on the predictive analysis of THMHH and SignalP could have led to incomplete results. Expanding evidence indicates that proteins are secreted through the unconventional protein secretion (UPS) function in diverse biological processes, including inflammation, development, immunity, and lipid metabolism [47]. For example, Drosophila glue (secreted through UPS) is used to protect fruit fly pupae from external environmental influences [48]. Zheng and Ge believe that UPS can be classified into two major types: vesicle-independent UPS and vesicle-dependent UPS [47]. Therefore, the remaining proteins may also be secreted proteins and participate in the interaction between leafhoppers and tea plants, which requires better screening methods for further research.

The enzymatic components of salivary proteins, a focal point in prior research, are considered highly valuable subjects for exploring salivary-induced plant defense responses. For instance, GOX (glucose oxidase) is known to act as an activator or effector in the oral secretions or saliva of various insects, including *Helicoverpa armigera* [49], *Helicoverpa zea* [50], *Ostrinia nuclealis* [51], and *Spodoptera exigua* [52]. Similarly, *β*-glucosidase has been identified to elevate the levels of host volatiles in the oral secretions of *Pieris brassicae* [53]. In the saliva of Hemiptera insects, cellulase in *Homelodisca vitripenis* [54] and endo-β-1,4-glucanase in *Nilaparavata lugens* [55] have been demonstrated as activators. Rice leafhoppers release watery saliva during feeding, which contains various substances, such as Ca^2+^-binding protein and enzymes, like laccase-1 and *β*-glucosidase [4]. Thus, researchers have speculated that the presence of trehalose, proteolytic enzymes, and other chemicals from tea green leafhoppers might play a crucial role in regulating the interactions between tea trees and leafhoppers [56]. Unexpectedly, none of the previously identified enzymes were discovered in our study. Studies have indicated that disulfide isomerase, an enzyme involved in protein folding, plays a role in folding salivary proteins in aphids [57]. Enzymes like ubiquitin-protein ligase, cooling coil domain-containing protein, and eukaryotic translation initiation factor 3, identified in this study, are associated with protein recognition and isomerization. However, despite their status as secreted proteins, the genes for these three proteins are not highly expressed in SG tissues. Further investigation is necessary to determine whether these proteins are indeed related to the folding of saliva proteins.

Plants deploy defense mechanisms to impede insect feeding by obstructing sieve tubes, and previous research has emphasized the crucial role of calcium channel-related proteins in facilitating this mechanism [21,58,59,60]. In our study, two proteins were identified: EfVDCC and EfRGN. Tissue-specific expression analysis revealed that *EfVDCC* was markedly overexpressed in the IN and FB tissues, while *EfRGN* showed specific overexpression in the testis and ovary tissues. This aligns with our findings, and a similar pattern was observed in Calexitin-1 isoform X1, which exhibited substantial expression exclusively in the testis and other tissues of white-backed planthoppers. Unlike the increased expression levels of Calexitin-1 isoform X1 in newly emerged female and male adults of white-backed planthopper [5], the expression levels of the *EfVDCC* and *EfRGN* genes were notably high during the first–third instar nymph stages. Therefore, further investigation is warranted to elucidate the specific roles of the EfVDCC and EfRGN proteins in inhibiting plant defense responses.

Piercing–sucking insects, reliant on sap from the xylem and phloem, face vulnerability due to imbalanced feeding structures. To meet their nutritional requirements, these insects depend on symbiotic bacteria or alternative pathways that release cellulose and other carbohydrate enzymes [61,62,63]. In the context of this study, the regulation of the vitamin distribution was attributed to the vitamin E transfer protein, also known as TTP. Fluorescence quantitative results indicated that the *EfTTP* gene exhibited significant and specific high expression levels in IN tissues, the first instar nymph stage, and female adults. This suggests the gene’s potential involvement in vitamin distribution during feeding. Given the differences between adult males and females, EfTTP emerges as a promising candidate for further exploration as a novel activator.

The mucin-like protein, being intricately associated with the formation of salivary sheaths, has consistently garnered attention in research endeavors. Prior studies have identified mucin-like proteins in the salivary secretions of brown planthoppers, small brown planthoppers, and white-backed planthoppers [2,5,6,14,64]. In the current investigation, two mucin-like protein genes, *Efmucin1* and *Efmucin2*, were also uncovered. The expression pattern of the *Efmucin1* gene aligned with reports from three planthopper species, exhibiting specific expression in SG tissue. This suggests that Efmucin1 may similarly contribute to the formation of saliva sheaths in tea leafhoppers. Mucin, characterized as a high-molecular-weight glycoprotein, shapes a mucin domain (MD) through O-linked glycosylation sites [65]. The MD sequence is rich in threonine, serine, and proline residues, typically forming tandem repeat sequences that are either identical or very similar [66]. Previous findings by Shangguan et al. [67] revealed that distinct mucin fragments induce different types of damage in tobacco plants. We hypothesize that the mucin peptide sequence may be linked to the process of plant cell identification and apoptosis. Considering that less than 25% of the mucin gene sequences across different species are homologous, the unique structure of mucin proteins could serve as a crucial factor for plants in identifying various pests.

In addition to providing nutrition and essential elements such as vitamins, phosphorus, sulfur, lipids, and amino acids for developing embryos, the lipid transporter protein Vg is believed to play a role in initiating defense responses in plants [59]. Vg has been identified in the watery saliva of three distinct planthopper species (brown planthoppers, small brown planthoppers, and white-backed planthoppers) and three distinct aphid species (potato aphids, wheat aphids, and pea aphids) [5,6,19,21,23,64,68]. The study also uncovered two Vg protein genes (*EfVg1* and *EfVg2*). Expression specificity tests revealed that the genes *EfVg1* and *EfVg2* had high levels of specific expression in the ovaries and FB tissues, with *EfVg1* also demonstrating some degree of specific expression in SG tissues. Ji et al. [69] found that small brown planthoppers could suppress rice defense responses by secreting VgC (C-terminal polypeptide of Vg) during feeding on rice. Interestingly, Zeng et al. [70] observed that brown planthoppers secreted VgN (N-terminal subunit of Vg) during feeding on rice, and VgN was also detected in the spawning fluid and on the surfaces of eggs. Furthermore, the VgN of brown planthoppers, small brown planthoppers, and white-backed planthoppers could all lead to an increase in jasmonic acid and jasmonic acid isoleucine, triggering rice defense mechanisms.

Moreover, this study unveiled, for the first time, three genes encoding the odor-binding protein (OBP) in the saliva proteins of insects in the order Hemiptera that had not been explicitly documented before. The identified OBP genes include *EfOBP1*, *EfOBP2*, and *EfOBP3*. Notably, all three genes exhibited higher expression levels in SG tissue compared to other tissues, with *EfOBP2*’s expression levels in SG tissue over 10 times greater than in others. Expression level analyses across various developmental stages revealed comparatively high expression levels during the first–third instar nymph stages. These findings strongly indicate that these proteins play a pivotal role closely associated with feeding. While limited research on OBPs in the saliva of bloodsucking insects has been published, *Anopheles stephensi* saliva has been reported to contain OBPs, primarily functioning as anticoagulants in the blood and aiding in blood sucking [71]. Research on *Anopheles gambiae* and *Aedes aegypti* has demonstrated that OBPs exhibit a significant affinity for biogenic amines, inhibiting the host’s inflammatory response and facilitating insect feeding [72,73]. Consequently, further investigation is warranted to elucidate the specific functions of OBPs in piercing–sucking insects.

## 5. Conclusions

This study identified 152 proteins across six major categories in saliva using LC-MS/MS. Employing bioinformatics techniques, 16 of these proteins were predicted to be secretory proteins. Ultimately, 20 genes encoding these putative salivary secretory proteins were successfully cloned. Tissue-specific and developmental stage-specific tests unveiled several potential elicitors or effectors, such as *EfVg*, *EfMucin*, and *EfOBP*, which could be associated with the feeding behavior of tea green leafhoppers and the defense responses initiated by tea plants.

## Figures and Tables

**Figure 1 insects-15-00296-f001:**
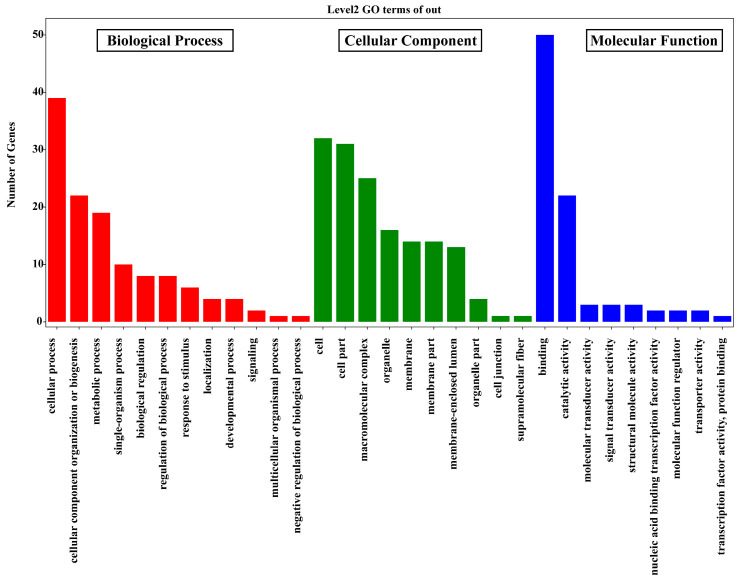
Gene Ontology classification of tea green leafhopper watery-saliva proteins. Saliva components were classified at the second level under three root GO domains: biological processes, molecular functions, and cellular components.

**Figure 2 insects-15-00296-f002:**
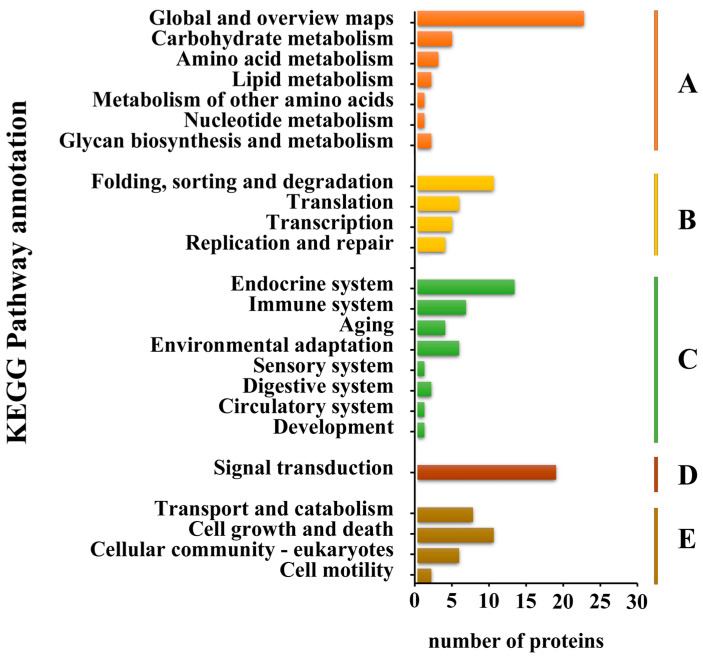
KEGG pathway classification of tea green leafhopper salivary proteins. The proteins according to the KEGG pathway involved were divided into five branches: A. metabolism; B. genetic information processing; C. environmental information processing; D. cellular processes; E. organismal systems.

**Figure 3 insects-15-00296-f003:**
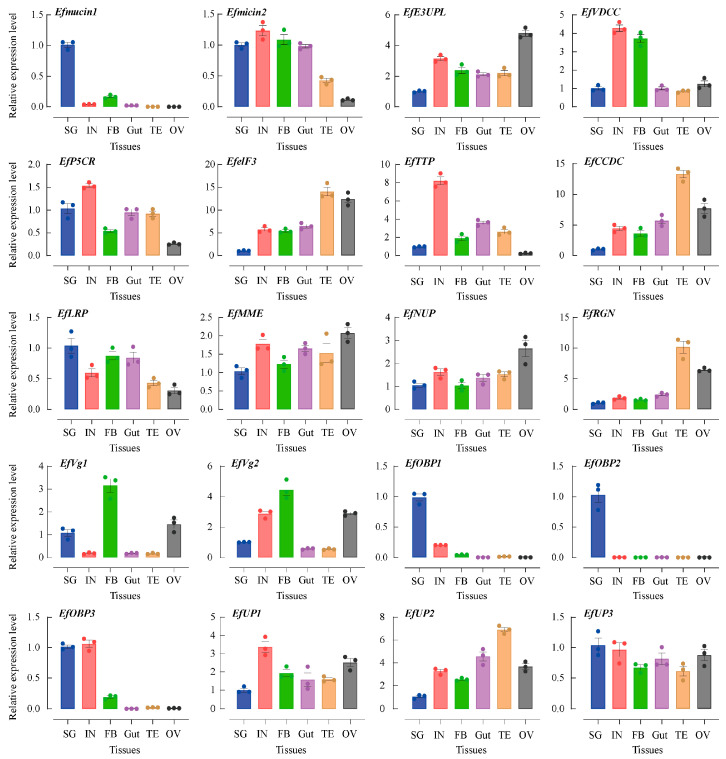
Tissue-specific expressions of *E. flavescens* genes encoding putative salivary secretory proteins. Gene expression analyses were examined through qRT-PCR using cDNA from five different tissues. SG, salivary gland; IN, integument; FB, fatty body; Gut, gut; OV, ovary; TE, testis. The EfTubulin and EfActin housekeeping genes were utilized to calculate the results using the 2^−ΔΔCt^ method.

**Figure 4 insects-15-00296-f004:**
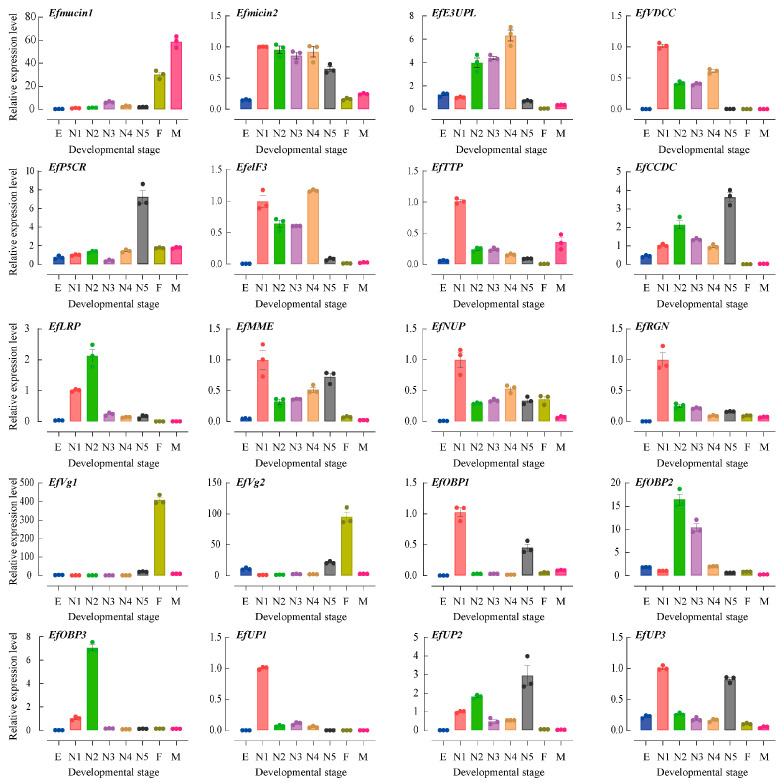
Developmental stage- and sex-specific expressions of *E. flavescens* genes encoding putative salivary secretory proteins. Gene expression analyses were examined through qRT-PCR using cDNA from various developmental stages and sexes, including eggs, 1st–5th instar nymphs, and recently emerged female and male adults. E, egg period; N1, N1-N5, 1st–5th instar nymphs, respectively; F, female adults; M, male adults. The EfTubulin and EfActin housekeeping genes were utilized to calculate the results using the 2^−ΔΔCt^ method.

**Table 1 insects-15-00296-t001:** Proteins identified in the tea green leafhopper salivary secretion with the putative secretion signal.

Gene	Accession No.	Annotation Gene Name	Sequence Length	ORF (aa)	Molecular Weight (kDa)	Completeness	Blastx Annotation
*Efmucin1*	OR504428	Mucin-like protein 1	4457	1392	154.03	Complete	mucin-2-like [*Hydra vulgaris*]
*Efmucin2*	OR504431	Mucin-like protein 2	6091	2010	208.62	Complete	mucin-2-like [*Halyomorpha halys*]
*EfE3UPL*	PP128337	E3 ubiquitin-protein ligase	10,845	3520	387.46	Complete	E3 ubiquitin-protein ligase HERC1 [*Halyomorpha halys*]
*EfVDCC*	PP128338	Voltage-dependent calcium channel	5694	1850	211.23	Complete	Voltage-dependent calcium channel type D subunit alpha-1 [*Cryptotermes secundus*]
*EfP5CR*	PP128339	Pyrroline-5-carboxylate reductase	885	292	31.09	Complete	pyrroline-5-carboxylate reductase [*Apis florea*]
*EfelF3*	PP128340	Eukaryotic translation initiation factor 3	1605	482	58.12	Partial	eukaryotic translation initiation factor 3 subunit A isoform X1 [*Nilaparvata lugens*]
*EfTTP*	PP128341	Alpha-tocopherol transfer protein	930	308	35.44	Complete	alpha-tocopherol transfer protein-like [*Nilaparvata lugens*]
*EfCCDC*	PP128342	Coiled-coil domain-containing protein	1407	465	53.75	Complete	coiled-coil domain-containing protein 47 [*Nilaparvata lugens*]
*EfLRP*	PP128343	Low-density lipoprotein receptor-related protein	1716	530	60.15	Partial	low-density lipoprotein receptor-related protein 4-like [*Priapulus caudatus*]
*EfMME*	PP128344	Membrane metallo-endopeptidase	2322	764	87.32	Complete	membrane metallo-endopeptidase-like 1 isoform X2 [*Bemisia tabaci*]
*EfNUP*	PP128345	Nuclear pore complex protein	4440	1478	154.44	Complete	nuclear pore complex protein DDB_G0274915-like [*Microplitis demolitor*]
*EfRGN*	PP128346	Regucalcin	843	160	17.71	Partial	regucalcin-like isoform X1 [*Nilaparvata lugens*]
*EfVg1*	PP128347	Vitellogenin1	6081	2021	223.745	Complete	vitellogenin [*Nephotettix virescens*]
*EfVg2*	PP128348	Vitellogenin2	5598	1792	199.335	Complete	vitellogenin [*Nephotettix virescens*]
*EfOBP1*	PP128349	Odorant-binding protein 1	546	142	16.22	Complete	odorant-binding protein 2 [*Rhyzopertha dominica*]
*EfOBP2*	PP128350	Odorant-binding protein 2	468	150	16.43	Complete	odorant-binding protein 8 [*Sogatella furcifera*]
*EfOBP3*	PP128351	Odorant-binding protein 3	438	136	14.74	Complete	general odorant-binding protein 19d-like [*Nilaparvata lugens*]
*EfUP1*	PP128352	Uncharacterized protein 1	2745	905	101.89	Partial	hypothetical protein B7P43_G00363 [*Cryptotermes secundus*]
*EfUP2*	PP128353	Uncharacterized protein 2	5562	1721	191.98	Complete	uncharacterized protein LOC111052746 [*Nilaparvata lugens*]
*EfUP3*	PP128354	Uncharacterized protein 3	693	228	25.62	Complete	putative OPA3-like protein CG13603 [*Onthophagus taurus*]

## Data Availability

Data is contained within the article or Appendix A.

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
