# Peer review of "Proteomic Analysis of Salivary Secretions from the Tea Green Leafhopper, Empoasca flavescens Fabrecius"

_insects, 2024, doi:10.3390/insects15040296_

Round 1

Reviewer 1 Report (Previous Reviewer 1)

Comments and Suggestions for Authors

The manuscript has been revised but there are some minor spelling errors. Fr example in Table 1:

A0A1B6CY97   towards the far right of the row, it should be transporter not "transpoter".

Please check the manuscript completely to eliminate other spelling errors.

Comments on the Quality of English Language

Minor spell check required.

Author Response

Dear reviewer:

Thank you very much for your patience and comments on our manuscript. We have revised our manuscript according to your comments: 

The manuscript was carefully checked and corrected by all co-authors. We highlight the changes we make in the manuscript by using the "Track Changes" function in Microsoft Word.

The detail as follows:

Comment 1: The manuscript has been revised but there are some minor spelling errors. For example in Table 1: A0A1B6CY97  towards the far right of the row, it should be transporter not "transpoter". Please check the manuscript completely to eliminate other spelling errors.

Response: Thanks for your kind suggestion. The manuscript have been checked and corrected by co-authors.

We would like to thank the referee again for taking the time to review our manuscript.

Reviewer 2 Report (Previous Reviewer 2)

Comments and Suggestions for Authors

This is a revised manuscript. 

Table 1 is not essential in the main text. I leave it to the authors, although the better place for Table 1 is in supplementary data.

For the description of the stage of the insets, newly emerged adult, the specific age needs to have further description, such as “less than 24 hr-old” or other precise description for the work that has been done.

Does three independent qRT-PCR mean three biological replications, three technical replications in the PCR, or in the RT-PCR?  Please specify what has been done.  If the qRT-PCR is without biological replication, it should be specifically mentioned to readers to set the reliability levels of the data.

Table 2 caption could be “Proteins identified in the tea green leafhopper salivary secretion with the putative secretion signal”.  However, as previous review mentioned, there are number of proteins that should not be secreted and/or not carrying the signal peptide in the N-termini.  For example, E3 ubiquitin-protein ligase should not be having secretion signal or a secretory protein.  Same for Voltage-dependent calcium channel and translation initiation factor.   I do not understand what process made authors include those genes in the table as the secreted proteins. Please fix this problem.

Inversely, I strongly think that the repertoire of the proteins in the salivary secretion indicates that the secretion mechanism is not only vesicle-mediated secretion but is also involving the apocrine type of secretion, which is the case of Drosophila salivary secretion at the time of pupation.  I suggest authors discuss this possibility and its importance. 

Author Response

Dear reviewer:

Thank you very much for your patience and comments on our manuscript. We have revised our manuscript according to your comments: 

The manuscript was carefully checked and corrected by all co-authors. We highlight the changes we make in the manuscript by using the "Track Changes" function in Microsoft Word.

The detail as follows:

Comment 1: Table 1 is not essential in the main text. I leave it to the authors, although the better place for Table 1 is in supplementary data.

Response: Thanks for your kind suggestion. Table 1 has been moved to supplementary data and named as Table S3.

Comment 2: For the description of the stage of the insets, newly emerged adult, the specific age needs to have further description, such as “less than 24 hr-old” or other precise description for the work that has been done.

Response: Thanks for your kind suggestion. We have added further description about newly emerged adult. ‘newly- emerged adult (less than 24 h old) tea green leafhoppers using a dissecting mi-croscope.’ The revised details can be found on Page 3, Lines 141 and 146.

Comment 3: Does three independent qRT-PCR mean three biological replications, three technical replications in the PCR, or in the RT-PCR?  Please specify what has been done.  If the qRT-PCR is without biological replication, it should be specifically mentioned to readers to set the reliability levels of the data.

Response: Thanks for your kind suggestion. We have revised the manuscript for ease of reading.

The details about hree biological replicates can be found on Page 3, Line 147 and Page 6, Line 270. ‘The three independent biological replicates of each samples were collected.’ and ‘Three independent qRT-PCR experiments were performed on total RNA samples. Mean and standard deviation values were calculated based on three independent biological replicates.’

Comment 4: Table 2 caption could be “Proteins identified in the tea green leafhopper salivary secretion with the putative secretion signal”.  However, as previous review mentioned, there are number of proteins that should not be secreted and/or not carrying the signal peptide in the N-termini.  For example, E3 ubiquitin-protein ligase should not be having secretion signal or a secretory protein.  Same for Voltage-dependent calcium channel and translation initiation factor.   I do not understand what process made authors include those genes in the table as the secreted proteins. Please fix this problem.

Response: Thanks for your kind suggestion. The caption of Table 2 has been modified. The name of salivary secretory protein in the manuscript has been modified to salivary putative secretory protein. The details about modification can be found on Page 16, Line 341.

Comment 5: Inversely, I strongly think that the repertoire of the proteins in the salivary secretion indicates that the secretion mechanism is not only vesicle-mediated secretion but is also involving the apocrine type of secretion, which is the case of Drosophila salivary secretion at the time of pupation.  I suggest authors discuss this possibility and its importance. 

Response: Thanks for your kind suggestion. We have added content on secreted proteins and necessary references in the discussion. The details about hree biological replicates can be found on Page 19, Line 449-461.

 ‘Although we screened 16 putative secretory protein from the repertoire of proteins for further investigation, relying solely on the predictive analysis of THMHH and SignalP can lead to incomplete results. Expanding evidence indicates that proteins secreted through unconventional protein secretion (UPS) function in diverse biological processes, including inflammation, development, immunity, and lipid metabolism [47]. For example, Drosophila glue (secreted through UPS) is used to protect fruit fly pupae from external environmental influences [48]. Zhang and Ge believe that UPS can be classified into two major types: vesicle-independent UPS and vesicle-dependent UPS [47]. Therefore, the remaining proteins may also be secreted proteins and participate in the interaction between leafhoppers and tea plants, requiring better screening methods for further research.’

This manuscript is a resubmission of an earlier submission. The following is a list of the peer review reports and author responses from that submission.

Round 1

Reviewer 1 Report

Comments and Suggestions for Authors

The manuscript "Proteomic analysis of salivary secretions ...Empoasca flavescens Fabrecius" reports on 152 salivary proteins identified by LC-MS/MS. The authors conducted a bioinformatic analysis of salivary secretions an predicted 16 secreted proteins which were cloned and transcriptionally analyzed in various tissues during different developmental stages. While the manuscript is of interest in terms of increasing our knowledgebase on insect saliva and how they contribute to insect plant interactions, there are several issues which would require to be addressed or clarified in the present version. I outline these concerns below:

1. The authors do not indicate when the saliva was collected. Were they collected during the feeding process and were the collections performed at the same time every day? There are bound to be changes in the secretome if the insects are not feeding on tea leaves and are only feeding on the 5% sucrose, in which case the design of the experiment would be flawed. Also, there are circadian changes in the salivary secretome - how did the authors know or account for these changes? What if the collections were performed at a time of day when such secreted proteins were not as abundant?

2. As in item 1 above, when were SGs, integuments, fat bodies, guts, testis and ovaries collected? What if the collection time does not capture the expression of genes when they would likely have been most expressed in a specific tissue?

3. 20 genes of secretory proteins were identified (Table 2) and the tissue specific gene expression was studied in SG, integument, Fat body, Gut, Testis and ovary (Fig.3). However, in Figure 3, SG does not seem to indicate the expression of any of these genes - the heat map indicates a complete blank for SG. Am I missing something? However, in the text the authors indicate that secretory proteins such as Efmucin1, EfOBP1, EfOBP2, EfOBP3, Efmucin2, EfLRP, EFVg1 and EfVg2 demonstrated peak expression levels in SG (Line 346-347). This contradicts the Figure S2.

4. In Lines 375-379 the authors indicate that certain genes showed peak expression levels in 1st nymphal instar, whereas Figure 4 N1 shows a complete blank heat map. This contradicts the Figure S3.

5. Line 269-270 indicates the authors performed relative quantitation for their qRT-PCR gene expression analysis. What was the expression relative to? There must be one tissue/ stage to which all other tissues/ stages are being compared - correct? That is what is relative quantitation.

I suggest the authors re-visit their data and create an appropriate heat-map. Also, For Figure S2 and S3, instead of mean with SEM/SD data, I would like to see the spread of individual data points.

Comments on the Quality of English Language

Some minor language and grammatical corrections are required.

Author Response

Dear reviewer 1:
Thank you very much for your patience and comments on our manuscript. We have revised our manuscript according to your comments:
The manuscript was carefully checked and corrected by all co-authors. We highlight the changes we make in the manuscript by using the "Track Changes" function in Microsoft Word.

The detail as follows:
Comment 1: The authors do not indicate when the saliva was collected. Were they collected during the feeding process and were the collections performed at the same time every day? There are bound to be changes in the secretome if the insects are not feeding on tea leaves and are only feeding on the 5% sucrose, in which case the design of the experiment would be flawed. Also, there are circadian changes in the salivary secretome - how did the authors know or account for these changes? What if the collections were performed at a time of day when such secreted proteins were not as abundant?
Response: Thanks for your kind suggestion. After 24 h, we collect sucrose solution (saliva-containing diet) fed to leafhoppers, which contains saliva components. Due to the small size of leafhoppers, it is difficult to collect saliva protein, which is not conducive to directly obtaining saliva. We have revised the description in the manuscript. The revised details can be found on Page 4, Lines 168-170.
After our preliminary experiments and observation of feeding behavior, leafhoppers generally have more feeding and activity behaviors from 5 a.m. to 9 a.m. and from 17 p.m. to 19 p.m., while other time periods are relatively quiet. Due to the time required for loading the device, it was chosen to complete the production of the saliva collection device within 1 hour after 12:00 am, and then every 24 hours after collecting saliva, the saliva collection device was remade to ensure that saliva from both feeding periods within 24 hours can be collected.
The main reason for choosing to collect saliva every 24 hours is because proteins are easily degraded in vitro, so saliva is collected every 24 hours for preservation.
Regarding experimental design, we also considered whether to directly obtain saliva from the host plant instead of from sucrose solution. However, due to the difficulty in raising the tea leafhopper, and the very small amount of saliva spit out by each leafhopper when feeding, the experimental results of the proteome will be greatly interfered with. Therefore, although this method has limitations, it is already a relatively effective method. We will strive to improve our feeding methods and experimental design for further in-depth and detailed research.

Comment 2: As in item 1 above, when were SGs, integuments, fat bodies, guts, testis and ovaries collected? What if the collection time does not capture the expression of genes when they would likely have been most expressed in a specific tissue?
Response: Thanks for your kind suggestion. We have supplemented the content in the manuscript. The revised details can be found on Page 3, Lines 141-145.
Due to the small size and immature tissue development of the 1-4th instar nymphs, it is not conducive to dissection. Through our preliminary experiments, we found that there is a strong feeding demand in the tea leafhopper before mating, and mating usually occurs 3-5 days after emergence. Therefore, we selected the newly-emerged adults for dissection.

Comment 3: 20 genes of secretory proteins were identified (Table 2) and the tissue specific gene expression was studied in SG, integument, Fat body, Gut, Testis and ovary (Fig.3). However, in Figure 3, SG does not seem to indicate the expression of any of these genes - the heat map indicates a complete blank for SG. Am I missing something? However, in the text the authors indicate that secretory proteins such as Efmucin1, EfOBP1, EfOBP2, EfOBP3, Efmucin2, EfLRP, EFVg1 and EfVg2 demonstrated peak expression levels in SG (Line 346-347). This contradicts the Figure S2.
Response: Thanks for your kind suggestion. We apologize for choosing an inappropriate image and color scheme. We have remade the images and used histogram with individual data. We have added a calculation method for data in the method.
Regarding the histogram, the 2−ΔΔCt values of the transcripts of the salivary secretory protein genes from various tea green leafhopper tissues samples were calculated and expressed as the fold change relative to expression of the SG tissues (expression = 1). The revised details can be found on Page 6, Lines 275-279.

Comment 4: In Lines 375-379 the authors indicate that certain genes showed peak expression levels in 1st nymphal instar, whereas Figure 4 N1 shows a complete blank heat map. This contradicts the Figure S3.
Response: Thanks for your kind suggestion. We apologize for choosing an inappropriate image and color scheme. We have remade the images and used histogram with individual data. We have added a calculation method for data in the method.
In the different developmental stage samples, the values were calculated and expressed as the fold change relative to expression of the 1st instar nymphs (expression = 1). The revised details can be found on Page 6, Lines 275-279.

Comment 5: Line 269-270 indicates the authors performed relative quantitation for their qRT-PCR gene expression analysis. What was the expression relative to? There must be one tissue/ stage to which all other tissues/ stages are being compared - correct? That is what is relative quantitation.
Response: Thanks for your kind suggestion. We have added a calculation method for data in the method.
In the different developmental stage samples, the values were calculated and expressed as the fold change relative to expression of the 1st instar nymphs (expression = 1). In the different developmental stage samples, the values were calculated and expressed as the fold change relative to expression of the 1st instar nymphs (expression = 1). The revised details can be found on Page 6, Lines 275-279.

Comment 6: I suggest the authors re-visit their data and create an appropriate heat-map. Also, For Figure S2 and S3, instead of mean with SEM/SD data, I would like to see the spread of individual data points.
Response: Thanks for your kind suggestion. We have added a calculation method for data in the method.
In the different developmental stage samples, the values were calculated and expressed as the fold change relative to expression of the 1st instar nymphs (expression = 1). In the different developmental stage samples, the values were calculated and expressed as the fold change relative to expression of the 1st instar nymphs (expression = 1). The revised details can be found on Page 6, Lines 275-279.

We would like to thank the referee again for taking the time to review our manuscript.

Reviewer 2 Report

Comments and Suggestions for Authors

Insects-2866065

The manuscript describes the results of proteomics of salivary secretion and transcriptomics of salivary glands of a pierce-sucking insect tea green leafhopper. The manuscript is descriptive without new discovery. Appropriate replications of the studies were not performed. Without major revision, the manuscript would not reach to the standard of this journal.

The style of English seems to be very casual, but not scientific.  For example, the “meticulous” operation of the experiment is not appropriate in scientific publication unless it explains what the measure for being meticulous is. There are multiple places saying “meticulously” prepared.

In line 21 and lines 35-36; the full names of the molecules need to be placed in the abstract. Abbreviations of the gene names could be spelled out.

Line 49’ “a lengthy coevolution process”, replace to “coevolutionary process” 

Lines 91 to 94; Authors have described previous findings.  However, the numbers are not clear whether those are all for secreted protein?  Or proteins in the salivary glands? For example, the 3603 is the number found in the transcriptomics and defined as potential elicitors, which is not comparable to the number 295 that was described for the number of proteins in the salivary sections.  

What is anatomical lens in line 141?  Does it mean stereo microscope?  I am not familiar with this term.

In material methods (line 177), Was the size fraction used for retained protein through a 10 kDa filter?  It is not clear whether the authors used the flowthrough or retained portion. 

Lines 275 to 278: This is a repeat of materials and methods except for the statement of identifying 152 salivary proteins.

Table 1 caption could be “Summary of proteins…”   I am not sure the lengthy 152 protein description can be in the main page.  Unless it contains a significant finding (which is the case as far as I read), it needs to move to the supplementary data.

Fig. 1. And 2 are also not informative and can move to supplementary data.

In the section 3.2., transmembrane domain is the signature for membrane integral protein, but not secretory protein.  Again, “meticulously” evaluated? What was meticulous? 

“christened”?  This is the first time that I've seen this term in a scientific manuscript.  Please simply say “named”.

Login number should be GenBank Accession number.

In table 2, some of these are apparently not predicted to be secreted.  Mis-categorization was apparently caused by misunderstanding of the transmembrane domain counted as the secretory protein as mentioned above. 

It is hard to understand whether the experiments, proteomics and tissue-specific and stage-specific qPCR, were repeated.  If not, at least three biological replications that allow statistical tests need to be done.

Comments on the Quality of English Language

The terms being used in scientific publication could be improved. 

Author Response

Dear reviewer 2:
Thank you very much for your patience and comments on our manuscript. We have revised our manuscript according to your comments:
The manuscript was carefully checked and corrected by all co-authors. We highlight the changes we make in the manuscript by using the "Track Changes" function in Microsoft Word.

The detail as follows:
Comment 1: The style of English seems to be very casual, but not scientific.  For example, the “meticulous” operation of the experiment is not appropriate in scientific publication unless it explains what the measure for being meticulous is. There are multiple places saying “meticulously” prepared.
Response: Thanks for your kind suggestion. We apologize for the ambiguity caused by the transition of synonyms. Scientific manuscripts should be rigorous, authentic, and accurate experimental content. We are very sorry for using imprecise and unscientific vocabulary. We have reviewed the manuscript and made corrections for any inaccurate vocabulary. Thank you very much for your rigorous and meticulous work attitude.

Comment 2: In line 21 and lines 35-36; the full names of the molecules need to be placed in the abstract. Abbreviations of the gene names could be spelled out.
Response: Thanks for your kind suggestion. we had already completed the modifications and spelled out their full names when these genes first appeared. The revised details can be found on Page 1, Line 21 and Page 37.

Comment 3: Line 49’ “a lengthy coevolution process”, replace to “coevolutionary process” 
Response: Thanks for your kind suggestion. “a lengthy coevolution process” was changed to “coevolutionary process”. The revised details can be found on Page 2, Line 50.

Comment 4: Lines 91 to 94; Authors have described previous findings.  However, the numbers are not clear whether those are all for secreted protein?  Or proteins in the salivary glands? For example, the 3603 is the number found in the transcriptomics and defined as potential elicitors, which is not comparable to the number 295 that was described for the number of proteins in the salivary sections.  
Response: Thanks for your kind suggestion. The numbers 295 and 3603 represent the total number of proteins identified in the SG transcriptome of Bemisia tabaci and Acyrthosiphon pisum, respectively, and are not salivary proteins or secreted proteins. We are very sorry for the confusion caused by our description of the differences between salivary protein, salivary secretory protein, and salivary gland secretory protein. We have revised the manuscript. The revised details can be found on Page 2, Line 91-97.

Comment 5:
What is anatomical lens in line 141?  Does it mean stereo microscope?  I am not familiar with this term.
Response: Thanks for your kind suggestion. “anatomical lens” was changed to “dissecting microscope”. The revised details can be found on Page 3, Line 142-143.

Comment 6: In material methods (line 177), Was the size fraction used for retained protein through a 10 kDa filter?  It is not clear whether the authors used the flowthrough or retained portion. 
Response: Thanks for your kind suggestion. We used a retained portion and the filtrate was discarded. We have supplemented the content of the manuscript. The revised details can be found on Page 4, Line 184-186.

Comment 7: Lines 275 to 278: This is a repeat of materials and methods except for the statement of identifying 152 salivary proteins.
Response: Thanks for your kind suggestion. We have removed duplicate content. The revised details can be found on Page 6, Line 285-286.

Comment 8: Table 1 caption could be “Summary of proteins…”   I am not sure the lengthy 152 protein description can be in the main page.  Unless it contains a significant finding (which is the case as far as I read), it needs to move to the supplementary data.
Response: Thanks for your kind suggestion. We still hope that the content of Table 2 can be retained in the main page of manuscript. Considering that this may be the first time the results of the tea leafhopper have been released, we hope that more people can conveniently see the identification results of the proteome, providing reference for subsequent research.

Comment 9: Fig. 1. And 2 are also not informative and can move to supplementary data.
Response: Thanks for your kind suggestion. We still hope that the content of Fig. 1 and 2 can be retained in the main page of manuscript. This is the information data that has been reprocessed, in order to facilitate the viewing of data results and facilitate the reading and understanding of the manuscript.

Comment 10: In the section 3.2., transmembrane domain is the signature for membrane integral protein, but not secretory protein.  Again, “meticulously” evaluated? What was meticulous? 
Response: Thanks for your kind suggestion. We are very sorry for using imprecise and unscientific vocabulary. We have reviewed the manuscript and made corrections for any inaccurate vocabulary.
In the section 2.6., The details can be found on Page 5, Line 240-243.
To predict the amino acid sequences of secreted protein, adherence to the criteria outlined in [35,37,38] was essential: (a) the presence of a signal peptide and absence of a transmembrane domain; (b) the presence of both a signal peptide and a transmembrane domain, with the latter falling within the range of the signal peptide.
We did not determine whether the protein contains a transmembrane domain to determine whether it is a secreted protein. This only provides a means to narrow down the scope of the research objective, in order to screen potential research subjects as soon as possible. In the end, our results also show that there are indeed some proteins that are secreted proteins, and they may be potential elicitors or effectors.

Comment 11: “christened”?  This is the first time that I've seen this term in a scientific manuscript.  Please simply say “named”.
Response: Thanks for your kind suggestion. “christened” was changed to “named”. We apologize for the ambiguity caused by the transition of synonyms. Scientific manuscripts should be rigorous, authentic, and accurate experimental content. The revised details can be found on Page 16, Line 341.

Comment 11: Login number should be GenBank Accession number.
Response: The login number in the table is derived from the GenBank Accession number.of National Biotechnology Information Center (NCBI). Partial sequence information has been released.

Comment 12: In table 2, some of these are apparently not predicted to be secreted. Mis-categorization was apparently caused by misunderstanding of the transmembrane domain counted as the secretory protein as mentioned above.
Response: Thanks for your kind suggestion. The proteins in Table 2 were screened according to the following criteria, as the results were predicted by bioinformatics and there may be some prediction errors in the results. So we conducted tissue-specific and developmental stage-specific experiments to further confirm whether it is a secreted protein and whether it is related to feeding.
In the section 2.6., The details can be found on Page 5, Line 240-243.
To predict the amino acid sequences of secreted protein, adherence to the criteria outlined in [35,37,38] was essential: (a) the presence of a signal peptide and absence of a transmembrane domain; (b) the presence of both a signal peptide and a transmembrane domain, with the latter falling within the range of the signal peptide.
We did not determine whether the protein contains a transmembrane domain to determine whether it is a secreted protein. This only provides a means to narrow down the scope of the research objective, in order to screen potential research subjects as soon as possible. In the end, our results also show that there are indeed some proteins (mucin, Vg and OBP) that are secreted proteins, and they may be potential elicitors or effectors.

Comment 13: It is hard to understand whether the experiments, proteomics and tissue-specific and stage-specific qPCR, were repeated.  If not, at least three biological replications that allow statistical tests need to be done.
Response: Thanks for your kind suggestion. We have revised the manuscript for ease of reading.
The details about proteomics can be found on Page 4, Line 164-167.
‘With twenty leafhoppers in each tube and no leafhoppers were reused for saliva collection, a total of 8000 individual leafhoppers were collected as one sample. Three independent biological replicates were harvested for each variety.’
The details about tissue-specific and stage-specific qPCR can be found on Page 6, Line 272.
‘Mean and standard deviation values were calculated based on three independent biological replicates.’

We would like to thank the referee again for taking the time to review our manuscript.